# Conformational dynamics of a nicotinic receptor neurotransmitter site

Mrityunjay Singh[1], Dinesh C Indurthi[2]*, Lovika Mittal[1], Anthony Auerbach[2]*, Shailendra Asthana[1]*

[1]Computational Biophysics and CADD Group, Computational and Mathematical Biology Center,Translational Health Science and Technology Institute, Faridabad, India; [2]Department of Physiology and Biophysics, University at Buffalo, State University of New York, Buffalo, United States

## eLife Assessment

This **useful** work provides insight into agonist binding to nicotinic acetylcholine receptors, which is the stimulus for channel activation that regulates muscle contraction at the neuromuscular junction. The authors use in silico methods to explore the transient conformational change from a low to high affinity agonist-bound conformation as occurs during channel opening, but for which structural information is lacking owing to its transient nature. The simulations indicating that ligands flip ~180 degrees in the binding site as it transitions from a low to high affinity bound conformation are **solid**. A limitation is the approximation of binding energies using Poisson-Boltzmann Surface Area and mismatch between calculated and experimental binding energies for two of the four ligands tested. Nonetheless, this work presents an intriguing picture for the nature of a transient conformational change at the agonist binding site correlated with channel opening.

**\*For correspondence:**
dinesh.indurthi@gmail.com (DCI);
auerbach.anthony@gmail.com (AA);
sasthana@thsti.res.in (SA)

**Competing interest:** The authors declare that no competing interests exist.

## Abstract

Agonists enhance receptor activity by providing net-favorable binding energy to active over resting conformations, with efficiency ($\eta$) linking binding energy to gating. Previously, we showed that in nicotinic receptors, $\eta$-values are grouped into five structural pairs, correlating efficacy and affinity within each class, uniting binding with allosteric activation (Indurthi and Auerbach, 2023). Here, we use molecular dynamics (MD) simulations to investigate the low-to-high affinity transition (L→H) at the Torpedo α−δ nicotinic acetylcholine receptor neurotransmitter site. Using four agonists spanning three $\eta$-classes, the simulations reveal the structural basis of the L→H transition where: the agonist pivots around its cationic center ('flip'), loop C undergoes staged downward displacement ('flop'), and a compact, stable high-affinity pocket forms ('fix'). The $\eta$ derived from binding energies calculated in silico matched exact values measured experimentally in vitro. Intermediate states of the orthosteric site during receptor activation are apparent only in simulations, but could potentially be observed experimentally via time-resolved structural studies.

## Introduction

Adult vertebrate nicotinic acetylcholine receptors (AChRs) are allosteric proteins that mediate synaptic transmission between motor neurons and skeletal muscle fibers. These pentameric ligand-gated ion channels are composed of four different subunits (2 α1, β1, δ, ε) arranged symmetrically around a central ion-conducting pore (*Rahman et al., 2020*; *Unwin, 1993*). Interactions between the neurotransmitter acetylcholine (ACh) and orthosteric sites in the extracellular domain (ECD) increase the probability of a global change in protein conformation that opens a distant (~60 Å) allosteric site (a 'gate') in the transmembrane domain (TMD), allowing cations to cross the membrane. In AChRs

the ligand-protein interactions occur sequentially in reactions called 'touch' (initial contact), 'catch' (agonist recognition), and 'hold' (receptor animation) (*Auerbach, 2024*; *Jadey and Auerbach, 2012*). Here, we use MD simulations to investigate the restructuring of a neurotransmitter site in hold.

The neurotransmitter sites switch from low- to high-affinity (L→H) at the start of the global gating isomerization, and the gate switches from closed to open at the end of the isomerization (C→O) (*Grosman et al., 2000*; *Purohit et al., 2013b*). We represent the overall gating conformational change of the receptor as $C_L \rightleftarrows O_H$, with capital letters indicating the functional status of the gate and subscripts indicating the functional status of the neurotransmitter sites.

Receptors change conformation (turn on and off) spontaneously, influenced only by temperature. In AChRs without agonists, a large energy barrier separates $C_L$ from $O_H$ so the probability of having the open-gate shape ($P_O$) is small (~$10^{-6}$) and the baseline current is negligible (*Jackson, 1986*; *Nayak et al., 2012*; *Purohit and Auerbach, 2009*). The higher affinity of $O_H$ compared to $C_L$ indicates that agonists provide extra binding energy to stabilize this state preferentially, increasing $P_O$ and membrane current. Importantly, agonists also stabilize the gating transition state (the separating barrier) to nearly the same extent as the end state, so the agonist-induced increase in $P_O$ is caused almost exclusively by an increase in the channel-opening rate constant. Consequently, neurotransmitters turn on AChRs rapidly to generate a fast-rising synaptic impulse.

Agonists are characterized by affinity, efficacy, and efficiency. Each orthosteric site has two affinities (agonist binding free energies), weak to L and strong to H. Their difference, H minus L ($\Delta G_H - \Delta G_L$), is the energy source that drives the otherwise unfavorable protein isomerization. This difference, plus the agonist-independent free energy of unliganded gating, determines the maximum of the dose-response curve (efficacy). This binding energy difference is the *amount* of free energy delivered to the gating apparatus at each orthosteric site.

A related, but distinct, agonist property is efficiency that, in contrast to efficacy, depends on the H/L binding free energy ratio, ($1 - \Delta G_L / \Delta G_H$) (*Nayak et al., 2019*). This ratio determines the *fraction* of binding free energy converted into local movements that stabilize the gating transition state and, hence, animate the protein. In a dose-response curve, efficiency is manifest as the correlation between the maximum response and $EC_{50}$ (*Indurthi and Auerbach, 2023*). Like affinity and efficacy, efficiency is a universal agonist attribute (*Auerbach, 2024*).

The salient energy changes at the orthosteric sites of adult-type muscle AChRs have been estimated experimentally by using electrophysiology (*Figure 1A*). These receptors have two approximately equal and independent neurotransmitter sites (*Nayak and Auerbach, 2017*). The total free energy change in the L→H transition for two neurotransmitters (–10.2 kcal/mol), added to the unfavorable unliganded isomerization free energy change (+8.3 kcal/mol at –100 mV), generates a favorable free energy change of gating (–1.9 kcal/mol) (*Nayak et al., 2012*). $\Delta G$ is proportional to the logarithm of the corresponding equilibrium constant by -RT, where R is the gas constant and T the absolute temperature (RT equals 0.59 at 23°C). Accordingly, the two independent L→H rearrangements add to increase the gating equilibrium constant by a factor of $5680^2$, to increase $P_O$ from $7.4 \times$ ~$10^{-7}$ to 0.96 (*Nayak et al., 2012*). Likewise, the L→H rearrangements increase the opening rate constant by almost the same factor, from ~$0.001$ s$^{-1}$ to ~$50,000$ s$^{-1}$. The result is a nearly perfect synaptic current that rises rapidly from almost zero to almost maximum (*Jackson, 1989*).

Agonist free energy changes in catch and hold determine affinity and efficacy. Our goal was to associate the $\Delta G_L \rightarrow \Delta G_H$ change in binding free energy associated with hold with local structural rearrangements at an AChR neurotransmitter site. MD simulations were used to explore conformational changes in the L→H transition at the α–δ neurotransmitter site of the *Torpedo* AChR (6UVW; *Rahman et al., 2020*). In brief, for four agonists L and H conformations were identified in the simulations, and binding free energies calculated approximately in silico matched those measured accurately (by using electrophysiology) in vitro. In L→H, three rearrangements were prominent for all agonists: (1) a rotation of the ligand about its cationic center ('flip'), (2) a downward displacement of loop C ('flop'), and (3) compaction, dehydration, and stabilization of the pocket ('fix'). Also, for all agonists a brief intermediate state connected the initial L and final H conformations. These results suggest that the movement of the agonist starts the global conformational change that ultimately opens the channel.

In a nutshell, four principles undergird the activation of AChRs by agonists. (1) With or without agonists, the receptor switches spontaneously and globally between resting and active conformations. (2) Agonists increase $P_O$ simply because they bind more strongly (with higher affinity) to the

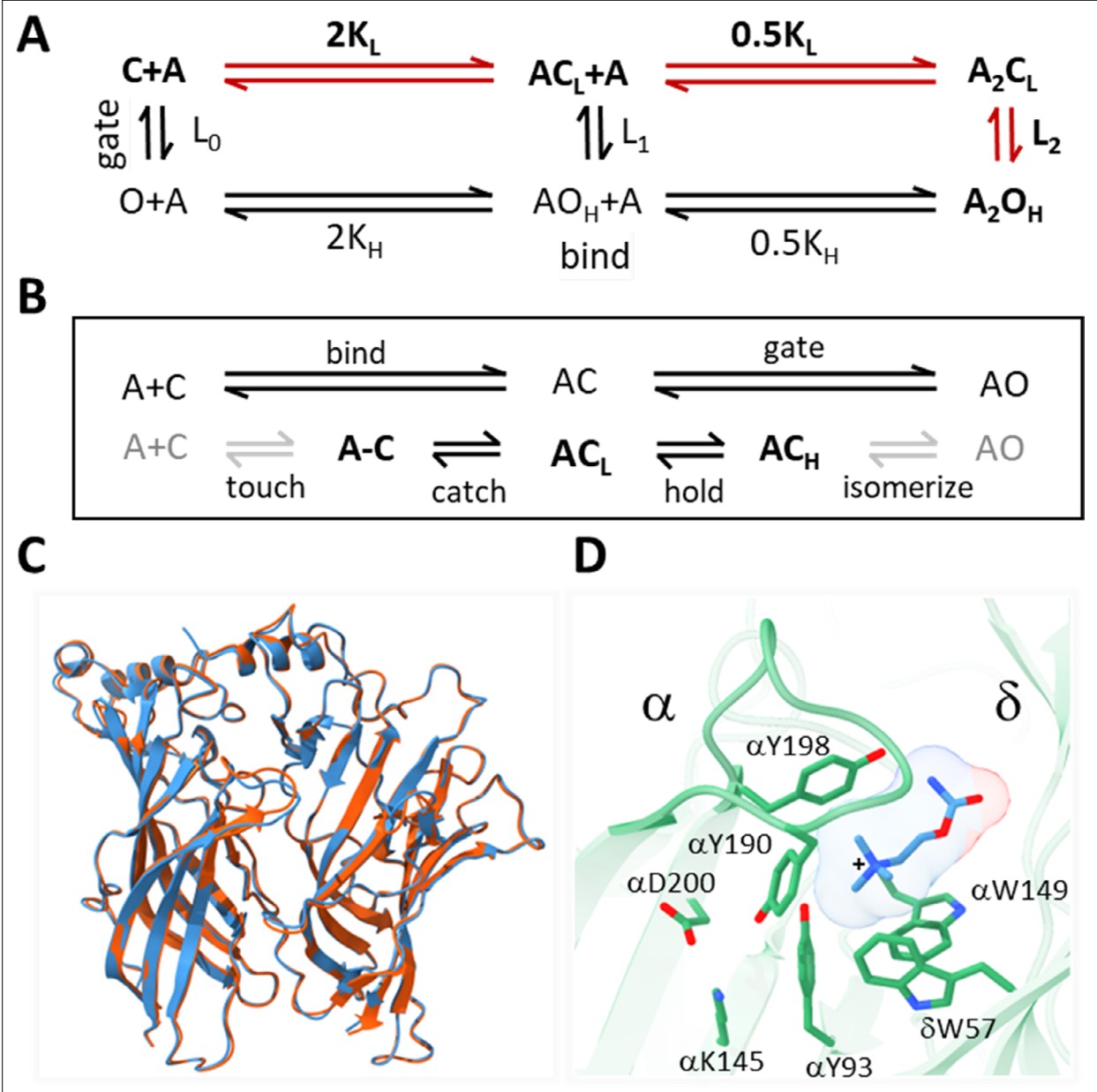

**Figure 1.** Acetylcholine receptor (AChR) activation. (**A**) Traditional scheme. Vertical is 'gating' and horizontal is 'binding;' red, the main physiological pathway. The isomerization between closed-channel/low-affinity ($C_L$) and open-channel/high-affinity ($O_H$) conformations occurs with or without agonists (equilibrium constant $L_n$; n, number of bound ligands), is spontaneous (depends only on temperature), and global (on a ~µs time scale). Agonists (**A**) bind weakly to $C_L$ equilibrium association constant $K_L$, free energy change $\Delta G_L$ and strongly to $O_H$ ($K_H$, $\Delta G_H$). The two orthosteric sites of adult AChRs are approximately equivalent and independent and there is no significant external energy, so by microscopic reversibility $L_2/L_0 = (K_H/K_L)^2$ (***Nayak and Auerbach, 2017***). (**B**) Expansions of binding (top, ends with catch) and gating (bottom, starts with hold). The agonist diffuses to and contacts the target ('touch') to form an encounter complex (A–C); a local 'catch' rearrangement establishes the low affinity complex ($AC_L$); a local 'hold' rearrangement establishes the high-affinity complex ($AC_H$); the remaining protein domains rearrange ('isomerize') without a further change in affinity, to generate a conducting channel ($AO_H$). Gray arrows, are steps that incur the same energy change for all agonists used in this study; black arrows, agonist-dependent free energy changes occur in catch ($\Delta G_L$) and in hold ($\Delta G_H$-$\Delta G_L$). (**C**) α-δ subunit extracellular domains; red, after toxin removal (6UWZ.pdb) and blue, apo (7QKO.pdb). There are no major deviations ($C_\alpha$ RMSD = 0.3 Å). (**D**) Closeup of the desensitized *Torpedo* α-δ subunit neurotransmitter site occupied by carbamylcholine (CCh, blue) (7QL6.pdb; ***Zarkadas et al., 2022***). In this is H conformation, three aromatic groups in the α subunit (149-190-198) surround the agonist's cationic center (+) together and provide most of the ACh binding energy (***Purohit et al., 2014***); the agonist's tail points away from the α subunit (*trans* orientation).

active conformation of their site. (3) Electrophysiology measurements show there is a linear free energy relationship at the heart of AChR activation by agonists: For families of agonists weak binding is a constant fraction of strong binding, regardless of affinity or efficacy. (4) MD simulations suggest that the first rearrangement in the switch from weak to strong binding is a pivot of the agonist about its cationic center.

## Results

### Hold

Traditionally, receptor activation is divided into distinct steps called 'binding' (formation of a ligand-protein complex) and 'gating' (receptor isomerization) (*Figure 1*). Here, we are concerned with their connection in the form of structural changes at a neurotransmitter site.

In AChRs, both binding and gating are aggregate processes (*Figure 1*). First, the agonist (A) diffuses to and contacts the resting target (touch), forming an 'encounter' complex, A-C. This step is approximately the same for small agonists. Then, a local protein rearrangement (catch) establishes the low affinity complex, $AC_L$. The expanded binding reaction scheme is $A+C \rightleftarrows A\text{-}C \rightleftarrows AC_L$ (*Jadey and Auerbach, 2012*). Gating involves sequential structural changes in receptor domains (*Gupta et al., 2017*; *Purohit et al., 2013b*). First, each neurotransmitter site undergoes another local rearrangement (hold) that establishes the high-affinity complex ($AC_H$), after which other domains restructure without a further change in agonist affinity, eventually opening the gate (isomerize). The expanded gating reaction scheme is $AC_L \rightleftarrows AC_H \rightleftarrows ... \rightleftarrows AO_H$ (*Figure 1B*). Hence, $AC_L$ is both the end state of 'binding' and the start state of 'gating'. Our interest here is the structural changes in the hold transition that links binding with gating, $AC_L \rightleftarrows AC_H$.

In AChRs, rearrangements that change agonist energy in catch ($\Delta G_L$) and in hold ($\Delta G_H - \Delta G_L$) are localized to the two orthosteric sites. Energy changes associated with the other events in the activation sequence, namely the chemical potential associated with the touch (*Phillips, 2020*) and downstream domain restructurings in the rest of the isomerization (*Gupta et al., 2017*) are approximately agonist-independent and not relevant to setting relative affinity, efficacy or efficiency. Although there are proposals and reports suggesting that protein-protein interactions at the ECD-TMD interface influence agonist binding energy, these are either unsupported (*Cymes and Grosman, 2021*; *Gupta et al., 2017*; *Nayak et al., 2012*) or in error (regarding N217K; *Purohit et al., 2015*). In AChRs, agonist binding energy is independent of energy changes outside (>~12 Å) the binding pocket (*Gupta et al., 2017*), so in our simulations we removed the TMD and studied only α−δ subunit ECD dimers.

The agonist's local energy changes at each orthosteric site in catch and in hold determine relative affinities ($\Delta G_L$ and $\Delta G_H$), efficacy ($\Delta G_H - \Delta G_L$), and efficiency ($1\text{-}\Delta G_L/\Delta G_H$). In AChRs, $\Delta G_L$ and $\Delta G_H$ have been measured experimentally by using electrophysiology for 23 agonists and 53 neurotransmitter site mutations (*Indurthi and Auerbach, 2023*), and these values served as a basis for identifying L and H structures in the MD simulations.

Apo, resting-C (with a bound toxin), and high-affinity desensitized AChR structures have been solved by using cryo-EM (*Rahman et al., 2022*; *Rahman et al., 2020*; *Zarkadas et al., 2022*). Desensitized (D) AChRs have a similar, perhaps identical, high affinity as $O_H$ (*Auerbach, 2020*; *Nayak and Auerbach, 2017*). Below, we report the results of MD simulations regarding the conformational dynamics that connect the end states of the hold rearrangement, $AC_L \rightarrow AC_H$ (*Figure 1B*, bottom), the event that triggers the isomerization that eventually opens the gate. Although neither is these states has been captured as a stable structure, the results suggest that they can be identified in MD simulations.

### Docking

We began by removing α-bungarotoxin from the 2.69 Å resolution structure of resting-C (6UWZ); (*Rahman et al., 2020*; *Figure 1C*) and docking agonists into the now-empty pocket. The docking simulations had strong binding scores ranging from –5.82 to –9.18 kcal/mol (*Figure 2A*), suggesting that the interactions were stable. However, unlike the agonist orientation apparent in cryo-EM structures of $D_H$ (*Noviello et al., 2021*; *Rahman et al., 2022*; *Rahman et al., 2020*; *Walsh et al., 2018*; *Zarkadas et al., 2022*; *Figure 1D*), the docked agonist's tail pointed towards (*cis*) rather than away

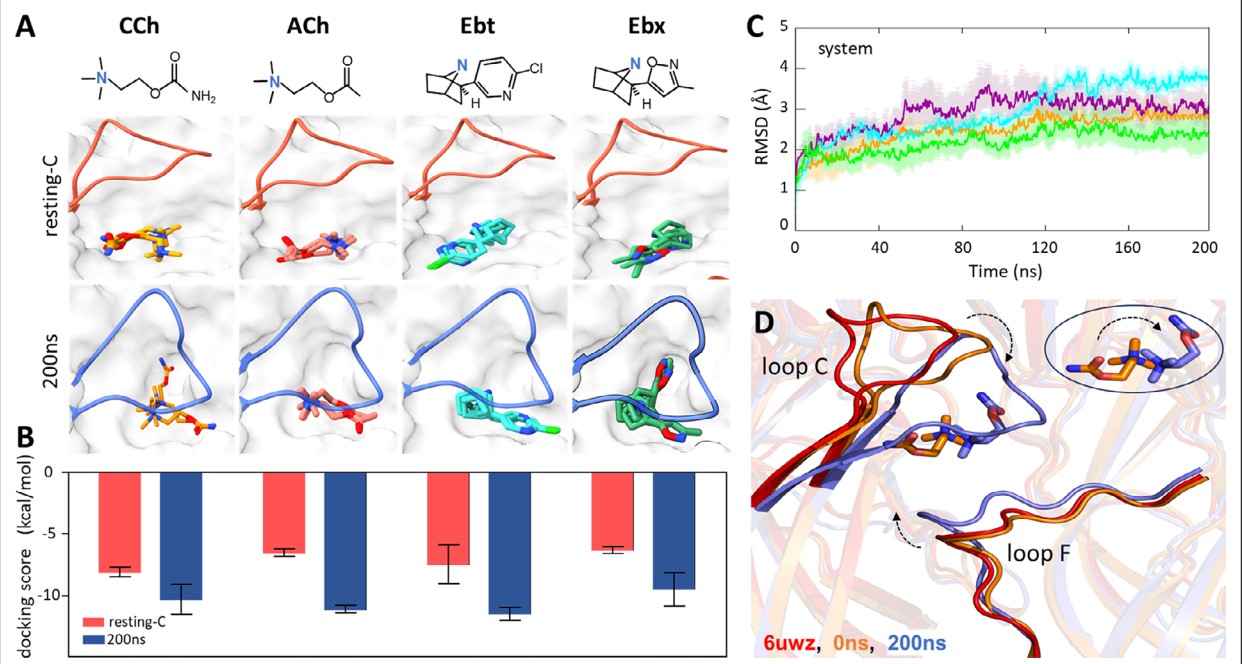

**Figure 2.** Agonist docking and loop dynamics. (**A**) Top, agonists (blue, cationic center): carbamylcholine (CCh), acetylcholine (ACh), epibatidine (Ebt), and epiboxidine (Ebx). Bottom, α–δ site with docked agonists (top three poses). Resting-C, 6UVW.pdb minus toxin (red): loop C is up and agonist is *cis*; 200 ns, after simulation and removal of CCh (blue): loop C is down and agonist is *trans*. (**B**) Bottom, *for all four agonists the* docking scores (mean ± SD, n=3) were more favorable after simulation. (**C**) $C_\alpha$ root-mean-square deviation (RMSD) (mean ± SD, triplicates) are stable after ~120 ns (ACh, cyan; CCh, green; Ebt, orange; Ebx, purple). (**D**) Close-up of the CCh-occupied pocket. Red, resting-C; orange, equilibrated (0 ns molecular dynamics, MD); blue, after 200 ns MD. IN the simulations, loop C flops down (arrow), loop F moves in, and the agonist flips *cis→trans* (circled inset).

The online version of this article includes the following source data and figure supplement(s) for figure 2:

**Source data 1.** Molecular dynamics (MD) simulation.

**Figure supplement 1.** Root mean square fluctuation (RMSF) of $C_\alpha$ for the extracellular domain (ECD) of α-δ subunits.

**Figure supplement 2.** RMSD of pocket residues during molecular dynamics (MD) simulations.

**Figure supplement 3.** Alignment of molecular dynamics (MD) simulations (carbamylcholine, CCh).

**Figure supplement 4.** Molecular dynamics (MD) simulation of the pentameric system.

**Figure supplement 5.** Conformational convergence between apo (blue) and with carbamylcholine (CCh) (green).

---

(t*rans*) from the α subunit (*Figure 2B*, top). This was a consistent result that pertained to the three top docking poses for all four ligands.

Because the *cis* orientation was not observed in any cryo-EM structure, we examined the top 200 docking scores for CCh. All poses were *cis*, with the tail pointing towards the α subunit and away from the main cluster of aromatic side chains. Apparently, in the unliganded 6UWZ and 7QKO structures (*Figure 1C*), the *trans* orientation observed in 7QL6 is unfavored. These results suggested that the agonist orientations in the end states are *cis* in $AC_L$ versus *trans* in $AC_H$.

### The L→H transition

We performed 200 ns MD simulations for each of the resting-C docked complexes, in triplicate (with different seed values). *Videos 1 and 2* show simulations with CCh on 2 time scales, expanded for the initial 200 ns and condensed for the full 1.0 µs. *Video 3* shows a 200 ns simulation with

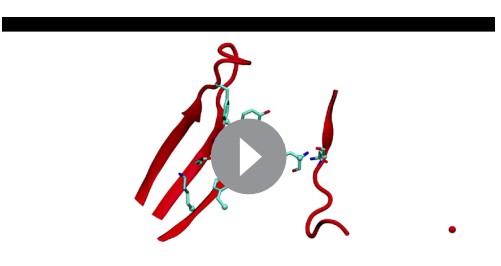

**Video 1.** Early Cis-to-Trans Transition of CCh in the AChR Binding Pocket.
https://elifesciences.org/articles/92418/figures#video1

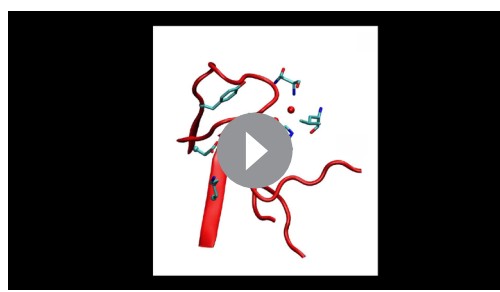

**Video 2.** Stabilization of agonist trans orientation and loop C displacement during the hold transition.
https://elifesciences.org/articles/92418/figures#video2

epiboxidine. The 200 ns simulations for all three replicates for all four agonists showed similar root-mean-square-deviation (RMSD) patterns with differences consistently below 1 Å (*Figure 2C*, *Figure 2—source data 1*) suggesting that the trajectories were sufficiently robust and reproducible. Although there were initial deviations, all systems became stable by ~120 ns. The root-mean-square-fluctuation (RMSF) of the $C_\alpha$ atoms showed flexibility around the pocket mainly in loops C and F (but not E) (*Figure 2—figure supplement 1*). The fluctuations at the base of the ECD were anticipated because the TMD that offers stability here was absent. However, there is no experimental evidence that the ECD-TMD interfacial region influences agonist binding energy, and (as described below) the match between experimental binding energies (whole receptors) and those calculated in the simulations (ECD dimers) further discounts this proposal. In addition, RMSD measurements show that, while agonists stabilize binding pocket residues (compared to apo; *Figure 2—figure supplement 2*), interfacial residues continue to fluctuate in the presence of agonists. The interfacial region plays an important role in setting the unliganded gating equilibrium constant ($L_0$) but not agonist affinity ($K_L$ or $K_H$) *Figure 3*, *Figure 4* .

In addition to the position changes of loops C and F during the simulations, in hold the ligand orientation inverted, *cis→trans* (*Figure 2D*). For all ligands and in all trajectories, at the start of the simulations, the agonist's tail points towards the α subunit and at the end, it points towards δ. This re-orientation represents a nearly complete somersault (a 'flip') about the agonist's main cationic center ($N^+$) that remains surrounded by the αW149-αY190-αY198 aromatic side chain cluster throughout (see *Figure 5*; *Figure 6—source data 1*). The final MD configuration (with CCh) aligns well with the CCh-bound cryo-EM desensitized structure (7QL6; RMSD <0.5 Å) (*Figure 2—figure supplement 3*). *Video 2* shows that the final, *trans* orientation with CCh remains constant for at least 1 μs, indicating that the structure had settled by 200 ns with no further major rearrangements. This result is consistent with functional measurements showing that D and O AChRs have indistinguishable affinities (citations in *Auerbach, 2020*).

Additional results from MD simulations support the ECD dimer system as sufficient to model the dynamics of the orthosteric site. First, the local rearrangements were similar in an ECD pentamer docked with ACh to those in the dimer, including the *cis→trans* flip of agonist (*Figure 2—figure supplement 4*). Second, 200 ns simulations of the apo ECD dimer show similar RMSF and RMSD patterns to those with ligand present (CCh) (*Figure 2—figure supplement 5A and B*). Third, in these simulations the downward displacement of loop C apparent with agonists (see below) was present, but less pronounced (*Figure 2—figure supplement 5C*).

To test whether rearrangements in the simulations served to enforce the *trans* pose, we removed the bound CCh from the final 200 ns MD structure and re-docked all four agonists. The preferred poses for all were *trans* (*Figure 2A*, bottom). In the simulation, the preferred agonist orientation switched from *cis* to *trans*. In addition, binding energies from docking scores were higher for the final 200 ns structures, consistent with an L-to-H transition (*Figure 2B*). Together, these results lead us to propose that MD simulations may indeed reflect actual rearrangements of a neurotransmitter site in the L-to-H, hold transition that turns on the AChR.

## Principal component analysis (PCA)

To identify the most prominent conformational states of the orthosteric site during the

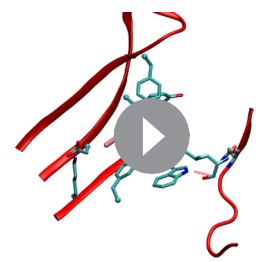

**Video 3.** Cis-to-Trans Transition of Ebx in the AChR Binding Pocket from L-to-H.
https://elifesciences.org/articles/92418/figures#video3

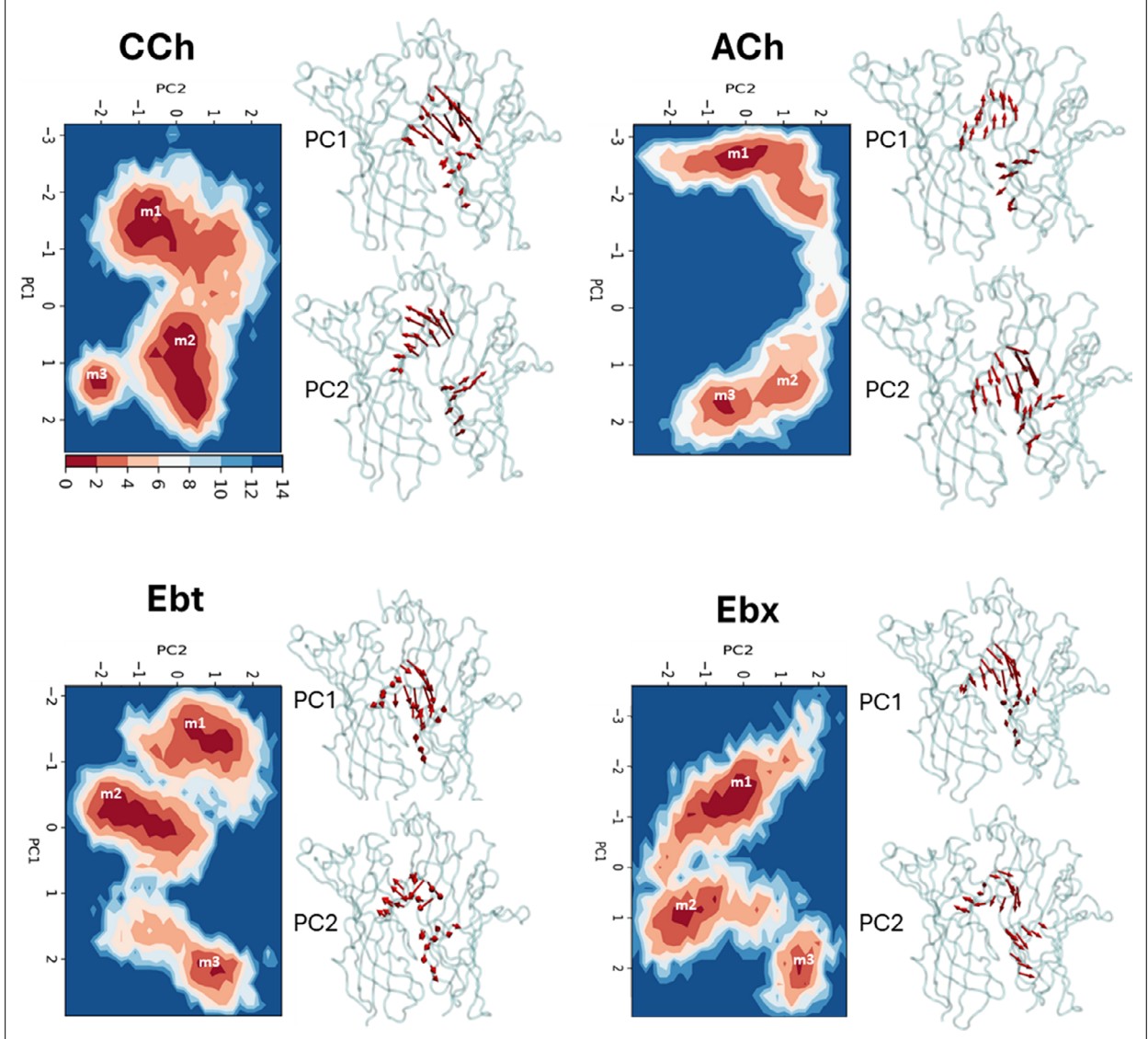

**Figure 3.** Principal component analysis (PCA). Left, for each agonist a plot of PC-1 versus PC-2, the first two principal components that capture the maximum variance in the trajectory (*Figure 3—source data 1*). Colors represent free energy value in kcal/mol (scale, upper left, bottom). For all agonists, there are three energy minima (darkest red) - m1, m2, and m3 - that correspond to different conformations of the neurotransmitter site. Right, 'porcupine' plots indicating that the direction and magnitude of changes PC-1 versus PC-2 is in loops C and F. From energy comparisons (*Figure 4*, *Figure 4—source data 1*) and temporal sequences (*Figure 3—figure supplement 2*, *Figure 4—source data 1*) we hypothesize that m1 represent state $AC_L$, m3 represents state $AC_H$, and m2 is an intermediate state in the $L \rightarrow H$, hold transition (*Figure 1B*).

The online version of this article includes the following source data and figure supplement(s) for figure 3:

**Source data 1.** Cumulative contribution of principal components (PCs) to the variance in molecular dynamics (MD) Simulations.

**Figure supplement 1.** Fraction of variance explained by eigenvalue rank for different simulation runs and conditions.

**Figure supplement 2.** Inner product heatmaps of principal components (PCs) from independent molecular dynamics (MD) simulations.

**Figure supplement 3.** Cluster analysis of ligand poses.

**Figure supplement 4.** Free energy landscapes as a function of PC1 and PC2.

**Figure supplement 5.** Ligand-protein Complexes during binding pose metadynamics (BPMD).

simulations, PCA was carried out based on the most significant fluctuations (principal components). Considering the first two principal components (PC-1 and PC-2) that captured the most pronounced $C_\alpha$ displacements (*Figure 3—source data 1*), backbone fluctuations revealed three energy minima we call m1, m2, and m3 (*Figure 3*). The first two principal components accounted for the majority of

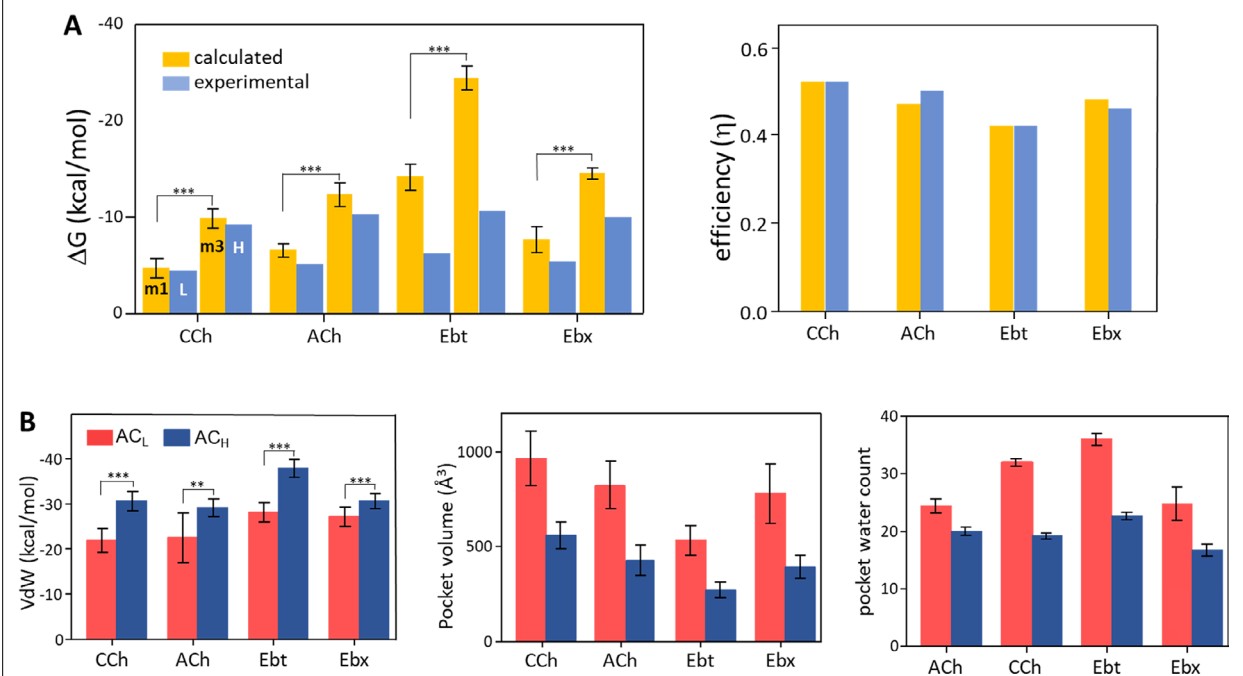

**Figure 4.** Binding free energies and pocket properties. (**A**) Calculated (yellow) versus experimental (blue) binding free energies for four agonists (structures in *Figure 2A*, top) (*Figure 4—source data 1*). Poisson-Boltzmann Surface Area (PBSA) calculations were done on clusters selected from m1 and m3 minima of principal component analysis (PCA) plots (*Figure 3*; *Figure 3—figure supplement 2*). Left, absolute ΔG, and right, efficiency (1-$\Delta G_L$/$\Delta G_H$). The agreement in efficiencies supports the hypothesis that m1 represents $AC_L$ and m3 represents $AC_H$ (**B**) In L→H (red→blue), VdW interactions (left) increase, pocket volume (center) decreases, and the number of water molecules in the pocket (right) decreases. Overall, the pocket stabilizes, compacts, and de-wets.

The online version of this article includes the following source data for figure 4:

**Source data 1.** Table of calculated and experimental binding energies (kcal/mol).

**Source data 2.** MM-Poisson-Boltzmann Surface Area (PBSA) components of the free energy calculation (see *Figure 4*).

the variance, as indicated by the rapid decrease in the fraction of variance with increasing eigenvalue rank (*Figure 3—figure supplement 1*). The consistency across different replicates for all four agonists supports the convergence of the PCA analysis. To further assess this consistency, we calculated the inner products of the top 10 principal components from each run. The resulting heatmaps reveal the degree of similarity between runs, with the inner product values of PC-1 and PC-2 reflecting considerable overlap (50–90%) in the principal dynamic modes captured in each simulation (*Figure 3—figure supplement 2*).

For all four agonists, all trajectories started in m1 and ended in m3, with m2 occurring as an intermediate (*Figure 3—figure supplement 3*). The free energy landscapes as a function of PC1 and PC2 (*Figure 3*, *Figure 3—figure supplement 4*) reveal variations in both the depths and variances of the wells across different agonists. The well depths, derived from -kTln $\rho$ values, reflect the frequency at which each state is occupied during the simulation relative to other states. This, and the variance that measures binding pose fluctuations around the mean positions for each minimum, provide insight into structural stability. In general, pronounced wells were present for all three minima for all four agonists (*Figure 3—figure supplement 4*). The only exception was m2 for ACh which was shallow, indicating a less stable intermediate state between m1 and m3. For all agonists, the m3 pose was more localized and stable compared to m1, with m2 being the least stable pose.

Pronounced fluctuations in loop C were evident in residue-wise RMSF plots and aligned well with the contributions of the first two principal components (PC-1 and PC-2) (*Figure 2—figure supplement 5D*). The apo- and CCh- occupied proteins adopt similar conformational spaces during the 200 ns simulation (*Figure 2—figure supplement 5E*). However, the presence of the agonist appears to increase conformational diversity, generating a broader distribution in the PCA plot. The bottom of the m1, m2, and m3 energy wells in *Figure 3* represent the most stable configurations of each

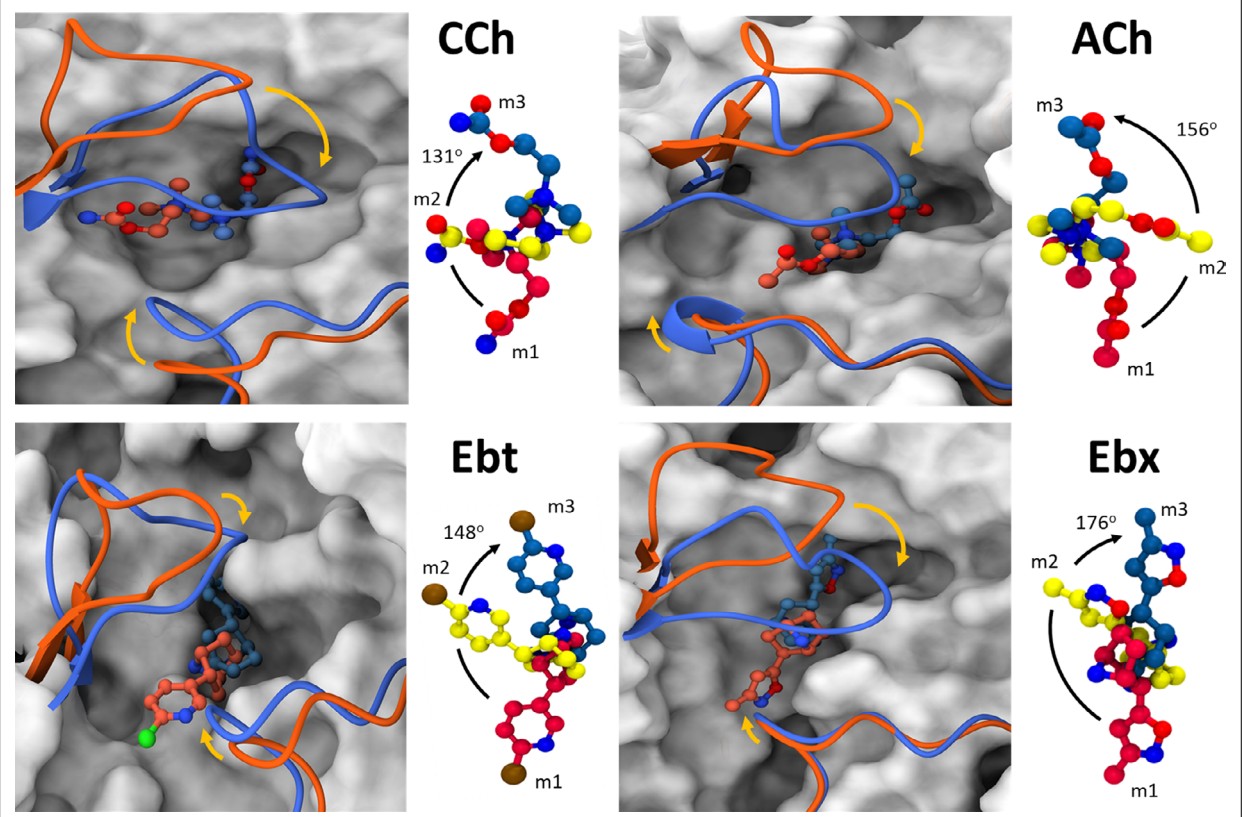

**Figure 5.** Agonist and loop movements in the hold (flip and flop). (**A**) Left, superimposed cartoons of AC_L (m1; orange) and AC_H (m3; blue). Loop C is upper left and loop F is lower right. In L→H (orange→blue) there is a *cis*→*trans* reorientation of the agonist (flip) and a downward movement of loop C (flop, arrow). R*ight*, agonist structure m1 (red) versus m2 (yellow) versus m3 (blue). Degree pertains to the m1→m3 pivot angle of the agonist.

The online version of this article includes the following figure supplement(s) for figure 5:

**Figure supplement 1.** Conformational change of the pocket cavity accommodates agonist re-orientation.

population and serve as reference points. In general (see next section), the ligand is *cis* m1 and *trans* in m3.

To further evaluate the stability of the binding poses in m1 and m3, we performed Binding Pose Meta Dynamics (BPMD), an enhanced sampling method for assessing ligand stability in solution. The BPMD calculations, which use CompositeScores derived from PoseScore (measuring ligand stability based on RMSD) and PersScore (evaluating the persistence of key ligand-protein interactions), were performed to assess stability. The BPMD scores consistently showed lower scores for m3 compared to m1, indicating higher ligand stability (*Figure 3—figure supplement 5*). Furthermore, the *trans* orientation in m3 was more stable than *cis* in m1, for all four ligands. Cross-docking of the *trans* orientation into the m1 conformation and the cis orientation into the m3 conformation confirmed that these are the preferred orientations. BPMD revealed that unstable ligand poses during the simulation are rarely occupied in the energy landscape and, therefore, make minimal contributions to binding free energy.

## Computed versus experimental binding energies

Although the well bottoms in *Figure 3* represent the most stable overall protein conformations, they do not directly convey information regarding agonist stability or orientation. To incorporate this information, we performed a cluster analysis of the ligand configuration using frames selected from the bottom of each PCA as well as inputs. The top three clusters, each having RMSD of ≤1.0 Å, shared a similar ligand orientation (*Figure 3—figure supplement 3*) and were selected to compute binding free energies (*Figure 4*, *Figure 4—source data 1*, *Figure 4—source data 2*). The remaining frames were disregarded. The fraction of frames from each minimum accepted for free energy calculations ranged

from 20% (ACh, m1) to 71% (Ebt, m3). On average, the fraction of frames selected was ~50%, showing the dynamic nature of the ligand in the pocket even in the regions of overall stability (*Videos 1 and 3*).

The above procedure defined populations of structures that might represent the end states of hold, $AC_L$ (m1), and $AC_H$ (m3), plus a previously unidentified intermediate state in the L→H transition (m2). We used PBSA (Poisson-Boltzmann Surface Area) computations of the selected structures for each agonist in each cluster population and compared the binding free energies calculated in silico with real-world values measured previously in vitro by using electrophysiology (*Indurthi and Auerbach, 2023*). PBSA is a computationally inexpensive method that provides a crude estimate of binding free energy (*Genheden and Ryde, 2015*; *Hou et al., 2011*). Importantly, the PBSA method was used only as a fingerprint for identifying states rather than for free energy estimation. We already knew actual binding free energies from the wet-bench experiments, so we used the PBSA values only to test the hypothesis that m1 corresponds to L ($\Delta G_L$) and m3 corresponds to H ($\Delta G_L$).

The PBSA and experimental ΔG values are compared in *Figure 4A* left (*Figure 4—source data 1*). For ACh and CCh, there was excellent agreement between $\Delta G_{m1}$ and $\Delta G_L$ and between $\Delta G_{m3}$ and $\Delta G_H$. The match was worse for the other 2 agonists, with the calculated values overestimating experimental ones by ~45% (Ebt) and ~130% (Ebx). However, in all cases, $\Delta G_{m3}$ was more favorable than $\Delta G_{m1}$, and the % overestimation compared to experimental values was approximately the same for m1 versus m3. Overall, the PBSA-electrophysiology comparison supported the hypothesis that m1 represents $AC_L$ and m3 represents $AC_H$.

Further support for these assignments comes from comparing calculated vs experimental efficiencies. Efficiency depends on the binding free energy *ratio*, $\Delta G_L/\Delta G_H$, rather than absolute ΔGs. As described elsewhere, efficiency is the agonist's free energy change in hold relative to its total free energy change in catch + hold (*Nayak et al., 2019*). It is the efficacy/high-affinity energy ratio, $(\Delta G_H-\Delta G_L)/\Delta G_H$ or $1-(\Delta G_L/\Delta G_H)$. Efficacy is the *amount*, and efficiency of the *fraction*, of agonist binding energy used to reduce the energy barrier separating L and H structures, to jumpstart gating.

*Figure 4A* right (*Figure 4—source data 1*) shows that efficiency values calculated in silico agreed almost perfectly with those measured experimentally. This result is strong support for the hypothesis that m1 represents $AC_L$ and m3 represents $AC_H$. Further, it indicates that efficiency can be computed accurately from structure alone. A possible reason for mismatches in ΔG but a match in efficiency is given in the Discussion.

Together, the docking scores, matches in free energies and efficiencies, and alignment of simulated m3 with cryo-EM desensitized structures support the hypothesis that MD simulations of a single pdb file faithfully reproduce the end states of the hold rearrangement. Hence, the MD trajectories likely reproduce the conformational dynamics of the orthosteric site in the initial stage of gating, $AC_L→AC_H$ (*Figure 1B*). This is the critical step that lowers the main barrier to activation of the allosteric site, to promote the otherwise unfavorable gating isomerization.

*Figure 4B* shows some structural parameters of the orthosteric pocket, L (**m1**) versus H (**m3**). All agonists showed an increase in Van der Waals (VdW) interaction energy in the hold rearrangement. This confirms the previous suggestion that L→H restructuring generates a smaller, more tightly packed pocket (*Tripathy et al., 2019*). The calculated (PBSA) increase was most pronounced with Ebt and least for Ebx, with CCh and ACh falling in between (*Figure 4—source data 2*). A compaction of the pocket is also apparent in *Figure 4C* that shows a reduction in binding pocket volume, and in *Figure 4D* that shows a decrease in the number of water molecules. As described below, the VdW contacts that form concomitant to the agonist flip and loop C flop establish a more compact, hydrophobic, and stable local environment.

## Flip-flop-fix

In L→H, the *cis→trans* pivot of the agonist was prominent in all simulations (*Figure 5*). On average, the ligands underwent a >130° pivot about the N⁺-αW149 fulcrum (*Figure 1D*). Also, in all instances, the flip began at the start of the simulation (m1).

A second consistent and prominent L→H restructuring event was the downward displacement (flop) of loop C toward the pocket center, defined as the $C_\alpha$ of αW149. This well-known 'clamshell closure' motion is conserved in related binding proteins (and in other receptors) and is thought to play a role in AChR activation by agonists (*Basak et al., 2020*; *Hansen et al., 2005*; *Hibbs et al., 2009*). In the simulations, loop C flop was measured as the displacement of its tip ($C_\alpha$ of αC192) in m3 relative

to its apo position. The distance traveled varied with the agonist (*Figure 6—source data 1*) but we do not discern a pattern in the extent of loop C flop with regard to agonist affinity, efficacy or efficiency.

In hold, residues that form the pocket cavity move, presumably to accommodate the reoriented ligand. In L, αY190, αY93, and δW57 are spaced apart, providing an open slot that accommodates the agonist's tail in the *cis* orientation (*Figure 5—figure supplement 1*). However, in H, loop C flop repositions αY190 closer to αY93, effectively filling the gap to create a unified surface.

In the desensitized H structure, αW149, αY190, αY198, and (to a lesser extent) αY93 and δW57A rings surround the agonist N$^+$ (*Figure 1D*). In adult-type mouse AChRs, substitutions of the 3 closest aromatic residues have somewhat different consequences. With ACh, deleting the ring (A versus W at 149, A versus F for 190 and 198) results in a loss (kcal/mol) of 3.0, 0.9, 2.1 for $\Delta G_L$ and 5.3, 2.8, 4.0

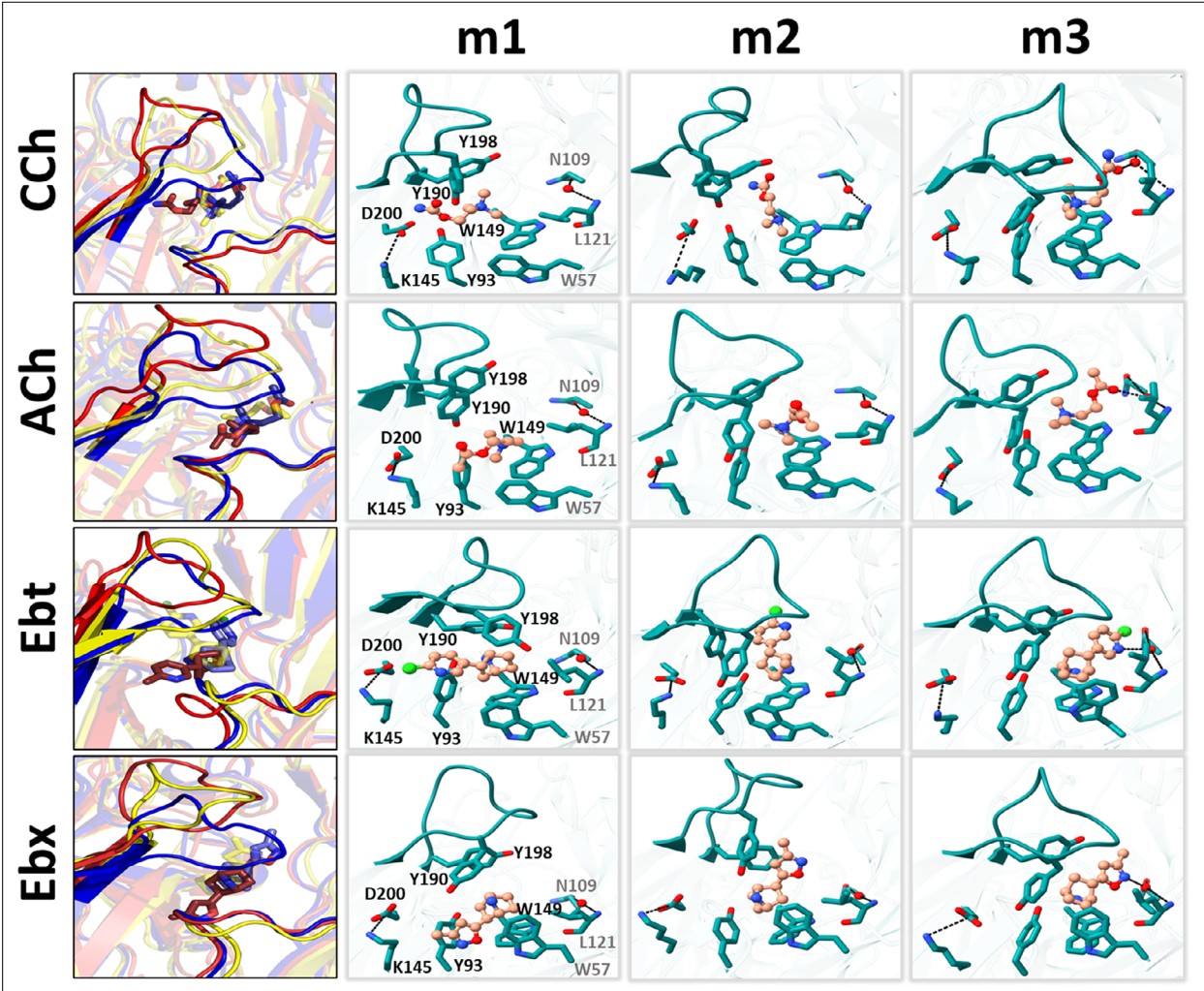

**Figure 6.** Representative snapshots in L→H (hold). *Left*, rearrangements of loop C, loop F, and the ligand (red, m1; yellow, m2; blue, **m3**); *right*, residue and ligand orientations. m1 is AC$_L$, m2 is an intermediate state, m3 is AC$_H$. (*Figure 1B*). In m1, a functional group in the agonist tail interacts with αY93 (all agonists) and αD200 (only CCh, and Ebt). The position and orientation of αW149 relative to N$^+$ of the agonist remains nearly unchanged m1→m2→m3 and serves as a fulcrum for the *cis→trans* flip (see *Figure 5*). In m2, the functional nitrogen at the agonist tail (CCh, Ebt, and Ebx) interacts with the hydroxyl group of αY198. For all ligands, αY190 repositioning and loop C flop (m1→m3) are correlated. In m3, the agonist fully flips to *trans*, facilitating VdW interactions, de-wetting, and the formation of water-mediated hydrogen bonds with the reactive group at its tail with δN109/δL121 backbone (loop E) *via* a structural water.

The online version of this article includes the following source data and figure supplement(s) for figure 6:

**Source data 1.** Residue distances.

**Figure supplement 1.** Binding cavities (α−δ subunit interface).

**Figure supplement 2.** Key residue distances over the course of molecular dynamics (MD) simulation.

for $\Delta G_H$, respectively (*Purohit et al., 2014*). αW149 (in loop B) is the main player in both L and H, but the biggest binding energy increase in hold comes from αY190 (in loop C). αW149 makes a cation-π bond with N+ that is important for ligand stabilization (*Xiu et al., 2009*; *Zhong et al., 1998*). In the simulations, the distance between these main aromatic groups and N+ decreases only slightly, L→H (*Figure 6*, *Figure 6—source data 1*). This aromatic cluster appears to act as a slippery anchor (a ball joint) that maintains the stability of the N+ position even as the tail rotates within it, approximately doubling agonist binding energy. The special role of αY190 is considered below.

In AChBP (perhaps L), the hydroxyl group of αY93 forms a hydrogen bond with the ligand's head group (*Celie et al., 2004*; *Hansen et al., 2005*). In the AChR L structure, the agonist's *cis* orientation appears to be stabilized by this hydroxyl that is close to a functional group on the agonist's tail. Likewise, the tail nitrogen in CCh, Ebt, and Ebx (amide or amine) forms a hydrogen bond with the hydroxyl group of αY198, but only in the intermediate m2 state. This interaction may guide the agonist pivot and provide transient stability. The absence of a secondary nitrogen in ACh possibly explains its alternate re-orientation route, as only this ligand was observed to flip counterclockwise (*Figure 5*). It may also explain why the m2 basin in the PCA plots for ACh is especially broad and shallow (*Figure 3*).

The L→H transition compacts, de-wets, and stabilizes the binding cavity (*Figure 4*; *Videos 1–3*). In H, the tail functional moiety (carbonyl or secondary ring amine) in the *trans* orientation is sandwiched between αC192, αC193, and αY198 (loop C), αT150 (loop B) and δC108, δN109, δL111, and δL121 (loop E) (*Figure 6—figure supplement 1*). As reported elsewhere (*Blum et al., 2010*; *Rahman et al., 2020*; *Zarkadas et al., 2022*), in our simulated H structure the functional group found in the tail of each agonist - the carbonyl group in CCh and ACh and the secondary ring amine in Ebt and Ebx - forms a hydrogen bond with the carbonyl of δN109 and amine of δL121 in the complementary δ subunit backbone *via* a structural water (*Figure 6*). The agonist's flip starts the m1→m2 transition, and the establishment of this water-mediated H-bond is completed in m2→m3 (*Figure 6*; *Video 2*). This water-mediated bonding and the increase in VdW interactions appear to both be crucial for stabilizing the loops and agonists in the HA conformation of the neurotransmitter site.

## Salt-bridge

Near the pocket, an αK145-αD200 salt bridge (*Figure 1D*) has been suggested to play an important role in receptor activation (*Beene et al., 2002*; *Gay and Yakel, 2007*; *Mukhtasimova et al., 2005*; *Padgett et al., 2007*; *Pless et al., 2011*). In the simulations, the rearrangement at αK145 is complex and agonist-dependent. In $AC_H$, this residue (i) makes an apparent contact with αY93, (ii) is approached by the αY190 hydroxyl group (*Figure 6—source data 1*), and (iii) loop F of the complementary subunit.

Interestingly, the mutation αK145A lowers the efficiencies of ACh and CCh (from 0.51 to 0.41) but has no effect on those of Ebt and Ebx (that remain at 0.41 and 0.46) (*Indurthi and Auerbach, 2023*). This agonist dependence in function has a parallel in structure. In the hold simulations, the αK145-αD200 distance (Å) increases from only 4.4 in apo to 6.1 or 5.1 in $AC_L$ (CCh or ACh). This spreading may relate to the agonist's tail being inserted into the αY93 slot in the *cis* orientation. In $AC_L→AC_H$, with ACh and CCh the αK145-αD200 separation (Å) shortens substantially to 2.8 or 2.6, making it more rigid. However, with Ebt and Ebx the αK145-αD200 distance remains unstable, fluctuating between 3–6 Å throughout the course of the simulation. Likewise, αK145-αY190 distance stabilizes at 5.3 Å or 6.1 Å in $AC_H$ with ACh and CCh but continues to fluctuate with Ebt and Ebx. In contrast, the αY190-αD200 distance remains stable in $AC_H$ for all ligands. This suggests that the instability in the αK145-αD200 and αK145-αY190 distances can primarily be attributed to αK145 (*Figure 6—figure supplement 2*).

## Discussion

AChRs are typical allosteric proteins (*Changeux, 2013*). Electrophysiology experiments indicate that agonists promote AChR activity by interacting with orthosteric sites in three stages, touch-catch-hold (*Auerbach, 2024*). We used MD simulations to probe structural changes in the hold rearrangement that animates the protein. By comparing binding energies calculated in silico with those measured in vitro we could identify provisionally in MD trajectories the end states $AC_L$ and $AC_H$, plus an intermediate configuration not detected in experiments. The conspicuous hold rearrangements discussed

below are a *cis→trans* pivot of the agonist (flip), an up→down displacement of loop C (flop), and decreases in pocket volume, water content, and loop fluctuations (fix).

Simulations require simplifications and are useful only insofar as they suggest experiments. Hence, we emphasize the rearrangements discussed below, are hypotheses that need to be tested experimentally. Nonetheless, agreements between cryo-EM (apo and desensitized) versus simulated (apo and $C_H$) structures, and between $\Delta G_L$ and $\Delta G_H$ free energies estimated from electrophysiology versus simulations, support our proposal that the simulated m1 and m3 structures reflect $AC_L$ and $AC_H$. At present, there are no time-resolved solved cryo-EM structures that correspond unambiguously to either of the hold end states, or to the intermediate m2 state. Absent these, we offer flip, flop, and fix as possibilities that can be tested.

There was reasonably good agreement between calculated and experimental absolute binding free energies for ACh and CCh (but not Ebt and Ebx), but the match was almost perfect for all ligands when free energy ratios (efficiencies) were compared (*Figure 4A*). In silico, the absolute values overestimated the experimental ones by approximately the same factor (different for each ligand) that, however, cancelled out in the ratio. We do not know the origin of this factor but speculate it could be caused by errors in ligand parameterization. For example, using the same value for the Born radius of nitrogen could overestimate $\Delta G$ more for ligands with azabicyclo versus quaternary ammonium groups. Regardless, the results show that agonist efficiency can be estimated in silico and, therefore, in the future could be a useful metric for analyzing dose-response curves and associating structure with function. Furthermore, excellent matches in energy measurements derived from pentamers studied via electrophysiology and those derived from ECD dimers in simulations, support the view that the ECD-TMD interface, absent in silico, is not an important determinant of agonist affinity. Rather, these matches are consistent with experiments that show that affinity is determined only by structural elements within ~12 Å of the ligand (*Gupta et al., 2017*), and that the AChR orthosteric sites operate independently (*Nayak and Auerbach, 2017*).

## Summary

MD simulations of the α−δ AChR neurotransmitter site indicate the following back-and-forth, generic sequence of rearrangements in $AC_L→AC_H$ (*Videos 1–3*).

1. $AC_L$ (m1): Agonist is *cis*, loop C is up, pocket is wide/wet/wobbly; agonist exits *cis*, starting a pivot about the $N^+$-αW149 fulcrum (flip); loop C starts to move down towards this fulcrum (flop).
2. Intermediate (m2): Agonist is ~half-rotated with tail stabilized by αY198; agonist completes the pivot to *trans* but with the tail not secured in the pocket; loop C moves down fully, to de-wet the pocket and deploy αY190 towards the salt bridge.
3. $AC_H$ (m3): Agonist is *trans*, the tail is secured by VdW contacts, an H-bond with structural water (fix); the pocket is narrow/dry/rigid.

Below, we consider the main structural elements associated with flip-flop-fix, namely the agonist, loop C, and the overall pocket, respectively.

## Agonist

With all agonists and in all simulations, the pivot of $N^+$ in a slippery, aromatic ball joint (αW149-αY190-αY198) starts the L→H transition. Similar movements of bound ligands have been identified in other proteins. In AMPA-type glutamate receptors, the neurotransmitter glutamate binds in either of two poses, crystallographic or inverted (*Yu et al., 2018*). In one MD simulation these interconverted, suggesting the bound ligand is not rigidly fixed (that is, T480-E705 might act as a slippery anchor). Although the alternative poses were not associated with experimental binding energies, the binding cleft is more closed in the crystallographic (presumably H) versus inverted (possibly L) orientation. Thus, it is possible that the inverse orientations of glutamate apparent in AMPA receptors are related to the flip of the agonist apparent in the AChR simulations.

Multiple ligand positions in binding pockets have also been reported in glycine receptors (*Yu et al., 2014*), 5-HT3 receptors (*Basak et al., 2020*), δ-opioid receptors (*Shang et al., 2016*), HIV-1 protease (*Klei et al., 2007*), and transthyretin (*Klabunde et al., 2000*). In enzymes, small bound substrate movements are apparent in hexokinase (*Bennett and Steitz, 1978*) and alcohol dehydrogenase (*Plapp, 2010*). Hence, it may not be unusual for the movement of a bound ligand to trigger an otherwise unfavorable local rearrangement to drive a change in protein function, either catalysis or

an isomerization (an 'induced fit;' *Richard, 2022*). Not only do untethered ligands have the ability to come and go, but they can also reorient to change binding energy and, hence, the local environment that couples to protein function.

In AChRs, the flip cannot pertain to all agonists. Some with high potency are perfectly symmetrical and completely lack a tail (tetramethylammonium). Although we have not yet identified in detail the specific dynamics that underpin $\Delta G_H$, additional factors beyond flip must contribute.

The PCA plots (*Figure 3*) show 3 populations, one of which (m2) occurs in time between m1 (AC$_L$) and m3 (AC$_H$) (*Figure 3—figure supplement 2*). In m2, the agonist's tail (CCh, Ebt, and Ebx) interacts with αY198, possibly to guide the clockwise pivot of the ligand and reduce the probability of a return to the *cis* orientation. The PBSA calculations indicate that $\Delta G_{m2}$ (that does not have an experimental correlate) is intermediate between $\Delta G_L$ and $\Delta G_H$ for ACh and CCh, but least stable for Ebt and Ebx (*Figure 4—source data 1*). We do not know the structural basis for this difference and its significance is not yet understood.

## Loop C

A number of previous observations place focus on the role of the downward displacement of loop C in receptor activation. In ACh binding proteins, the extent correlates with agonist potency (*Basak et al., 2020*; *Du et al., 2015*; *Polovinkin et al., 2018*) even if voltage-clamp fluorometry of other receptors shows it to be similar to agonists and antagonists (*Chang and Weiss, 2002*; *Munro et al., 2019*; *Pless and Lynch, 2009*).

In our simulations of the apo AChR (without an agonist), loop C flops partially to narrow the aqueous connection between the bath and the pocket (*Figure 2—figure supplement 2C*). It is, therefore, interesting that electrophysiology measurements show the agonist association rate constant ($k_{on}$) is substantially *faster* (approaching the diffusion limit) and less correlated with potency to H versus L conformations (*Grosman and Auerbach, 2001*; *Nayak and Auerbach, 2017*). That is, it appears that loop C flop generates a *higher* affinity, but not by restricting the rate of agonist entry. All four agonists dock in the *cis* orientation to L (wide entrance) versus the *trans* orientation to H (narrow entrance) (*Figure 2B*). Agonists indeed bind weakly to *cis* versus *trans*, so it appears that a structural element(s) disfavors adopting the *trans* pose when loop C is up. We speculate that in addition to narrowing the entrance, the loop C flop and accompanying rearrangements serve to reduce this inhibition and allow the *trans* orientation. To test this hypothesis, both the encounter complex (the starting state of catch) and possible inhibiting structural elements need to be identified.

The movement of loop C toward the pocket center resembles the 'clamshell closure' apparent at other receptor binding sites (*Chen et al., 2017*; *Hansen et al., 2005*; *Hibbs and Gouaux, 2011*). However, in light of the $k_{on}$ experimental results mentioned above, using 'closure' and 'capping' to describe this motion is misleading because these words imply placing a lid (a steric barrier) over the pocket that would be expected to reduce, not increase, the agonist association rate constant. Despite the fact that loop C flop narrows the entrance and is correlated with higher affinity, electrophysiology experiments indicate clearly that it does not restrict agonist entry. For this reason, we prefer the functionally neutral words 'flop' and 'up/down' to describe loop C displacement and position.

Truncation of loop C eliminates activation by ACh but has almost no effect on unliganded activation (*Purohit and Auerbach, 2013a*). Interestingly, leaving only loop C residues αY190 and αY198 in place retains activation by ACh. Apparently, repositioning of one (or both) of these aromatic side chains that anchor N+ in the up→down displacement of loop C is sufficient to maintain activation by the neurotransmitter, but not necessary for constitutive activity. Loop C flop may be essential for activation by agonists but not for constitutive activation.

An Ala substitution of αY190 results in a large loss of ACh binding energy with about half coming from the removal of the hydroxyl group (*Purohit et al., 2014*). In contrast, the mutations αY198F and αY93F have essentially no effect on ACh binding energy. These results suggest that repositioning of αY190 as a result of loop C flop is important for activation by ACh (*Figure 6*). The αY190 ring fills the slot that opens next to αY93 when the agonist exits *cis*, preventing a return (*Figure 5—figure supplement 1*).

Loop C flop positions the αY190 hydroxyl close to the αD200-αK145 salt bridge. It has been proposed that this rearrangement triggers the AChR allosteric transition (*Beene et al., 2002*; *Gay and Yakel, 2007*; *Mukhtasimova et al., 2005*; *Padgett et al., 2007*; *Pless et al., 2011*). In support of

this interaction, the mutation αK145A reduces ACh (and choline) binding energy substantially but has no effect when the αY190 hydroxyl is absent (*Bruhova and Auerbach, 2017*). Also, in the simulations with ACh the downward displacement of loop C positions this hydroxyl far (~5.0 Å) from αD200 and αK145 (*Figure 6—source data 1*). Although no correlation was found between these separations and the degree of loop C displacement with different agonists, the gap was smaller with high- (ACh and CCh) versus low-efficiency agonists (Ebt and Ebx). We sought, but did not find, structural water in this gap. The αY190 hydroxyl is an important determinant of L and particularly H binding energy, but the simulations have not revealed the mechanism.

In summary, the simulations confirm an αY190 hydroxyl-bridge interaction but do not reveal its nature or consequence. Rather than triggering the gating transition, the αY190-bridge interaction may serve mainly to stabilize the down position of loop C and, hence, the H pocket. Substrate-induced loop displacements are known to deploy residues to enhance catalysis, for example in chymotrypsin (*Ma et al., 2005*) and triosephosphate isomerase (*Brown and Kollman, 1987*; *Nickbarg et al., 1988*). Repositioning αY190 by loop C flop could be analogous.

## Pocket

Distances of the aromatic-$N^+$ anchor remain relatively constant in the hold transition (*Figure 6*, *Figure 6—source data 1*). We hypothesize that either very small distance changes associated with cation-π bonds are important, or that some of the extra favorable agonist binding energy in H is generated somewhere else. For the agonists we examined, the pivot places the tail into the main binding cavity (*Figure 6—source data 1*; *Video 1*) with *trans* orientation secured by VdW interactions (dehydration), a hydrogen bond with a structural water bonded to the loop E backbone of the complementary subunit, and, possibly, the αY190-salt bridge interaction. The net result to the overall pocket is a reduction in volume (*Figure 4C*) and an increase in the stability of loops C and F (*Figure 3*).

In the simulations, the first two structural changes, distributed over the large surface area of the cavity, were robust and consistent for all four agonists. Compact, hydrophobic binding pockets are associated with functional activation in other ion channels (*Furukawa et al., 2005*; *Sigel and Stein-mann, 2012*) and observed in other proteins including β2-adrenergic receptors, opioid receptors, and HIV-1 protease (*Huang et al., 2015*; *Rasmussen et al., 2011*; *Wlodawer and Erickson, 1993*). A future objective is to quantify each source of increased binding energy with regard to the aromatic cluster, αY190 salt-bridge interaction, general pocket restructuring, and H-bond with structural water.

## After hold

The next step In AChR gating is a rearrangement in the extracellular domain. In related receptors, upon activation this domain becomes more compact and stable, a rearrangement called 'unblooming' (*Sauguet et al., 2014*; *Taly et al., 2009*). This restructuring echoes that of hold at the AChR neurotransmitter site. The L→H rearrangement of the pocket may nucleate the subsequent restructuring of the extracellular domain through currently unknown connections.

In the complementary δ subunit (*Figure 1D*), loop E (β5-β6 strands) has been implicated in ligand discrimination (*Basak et al., 2020*; *Muroi et al., 2009*; *Pless and Lynch, 2009*). The double mutation F106L+S108 C in the β2 subunit of the α4β2 nicotinic AChR subtype significantly decreases the action of Ebt without altering that of ACh (*Tarvin et al., 2017*). In addition, loop E residues are responsible for the binding energy differences α−γ versus α−δ (*Nayak et al., 2016*) and reduce site instability caused by loop C mutations (*Vij et al., 2015*). Although our simulations do not shed light on these matters, the temporal sequence flip-flop-fix suggests that perturbation of the loop E backbone could also play a role in the downstream activation of the receptor.

Agonists belong to efficiency classes in which $\Delta G_L/\Delta G_H$ is the same for all members. So far five such classes have been identified experimentally in adult-type AChRs, indicating that there also are five pairs of $AC_L/AC_H$ structural classes (*Indurthi and Auerbach, 2023*). Although in the simulations loops C and F movements and salt bridge distances were agonist dependent, with only four ligands we cannot associate any of these rearrangements with efficiency classes. Additional simulations using more agonists and with binding site mutations may provide better understanding of the dynamics underpinning molecular recognition in catch and protein animation in hold, and potentially reveal the structural basis of their linkage (*Auerbach, 2024*).

## Implications

Several aspects of this work may have general implications. The L and H end states of hold emerged from simulations of a single protein structure and were identified by comparing experimental with binding energies calculated using an inexpensive, approximate method. In general, this approach could be useful because at present cryo-EM structures of the short-lived $AC_L$ states are difficult to obtain. Agonist efficiency was calculated accurately from structure alone. This, too, could be useful because efficiency allows affinity and efficacy to be calculated from each other (*Indurthi and Auerbach, 2021*). Using MD to estimate efficiency could facilitate dose-response analysis and speed drug discovery. Finally, the simulations suggest that a rotation of the agonist is the trigger that eventually turns on the receptor. While it makes sense that the movement of a newly added structural element (the signaling molecule) starts the receptor's global conformational change, this hypothesis should be tested experimentally.

# Materials and methods

## Hardware and software

All computational studies were carried out on a Linux Ubuntu 20.04-based workstation with 126 GB RAM, 64 CPU cores, and two RTX A5000 24 GB GPU cards, as well as an Ubuntu 20.04-based server equipped with 132 GB RAM, 96 CPU cores, and two RTX A5000 24 GB GPU cards. The software utilized includes the Maestro release 2022–2 graphical user interface (GUI) of the Schrödinger software suite (BioLuminate, Schrödinger, LLC, NY, USA), Amber22 (*Case et al., 2005*), VMD 1.9.3 (*Humphrey et al., 1996*), ChemDraw Professional 15.1 (*Purushotham et al., 2022*), ChimeraX (*Pettersen et al., 2021*), and PyMOL (PyMOL Molecular Graphics System, Version 2.0, Schrödinger, LLC).

## Protein preparation

Agonist energy changes are approximately independent of energy changes in the protein beyond the orthosteric pocket. In the simulations, we removed the transmembrane domain and simulated only $\alpha-\delta$ subunit extracellular domain dimers. The $AC_L$ starting structure of the hold transition was resting C (PDB: 6UWZ) after toxin removal that is essentially the same as the apo structure (*Zarkadas et al., 2022*; *Figure 1C*). The 4 agonists we examined were acetylcholine (ACh), carbamylcholine (CCh), epibatidine (Ebt), and epiboxidine (Ebx) (*Figure 2A*). These are of similar size (MW 187, 183, 209, and 178, respectively) but have different affinities, efficacies, and efficiencies (0.50, 0.52, 0.42, and 0.46, respectively) (*Indurthi and Auerbach, 2023*).

Protein preparation was conducted using the Protein Preparation Wizard (PPW) (*Purushotham et al., 2022*) and OPLS4 force field. Bond orders were assigned to atoms based on the CCD database, with zero bond orders given to bonded metals. Missing hydrogen atoms were inserted, and disulfide bonds were placed appropriately. During preprocessing, structural refinement was performed with Maestro's Prime utility, optimizing and adding any missing side chains and residues. Missing residues were formulated by combining SEQRS information from the PDB files and optimized using Prime's ePLOP module. The ionization state of all heterogeneous groups, including ligands, bound metals, and charged amino acids, were determined using the EpiK tool at a pH range of 7.0 ± 2.0. Subsequent force field minimization allowed for water sampling and basic restraint minimization, during which only hydrogen atoms could be freely reduced.

The hydrogen bonding (H-bond) network underwent significant modification. The terminal chi angle for residues Asn, Gln, and His was sampled through 180 degree flips, altering their spatial H-bonding capabilities without affecting the fit to electron density. The neutral and protonated states of His, Asp, and Glu, as well as two His tautomers, were sampled. Hydrogens on hydroxyls and thiols were also analyzed to enhance the H-bond network. Finally, the ProtAssign method was employed in two modes: a 'regular' mode, which was completed in seconds, and an 'exhaustive' mode that considered many more states and could take minutes or hours, depending on the H-bond network complexity.

## Ligand preparation

The two-dimensional SDF structures of CCh, ACh, Ebt, and Ebx were retrieved from the PUBCHEM database. These agonists were chosen because they have a similar size and represent different

efficiency classes (*Indurthi and Auerbach, 2023*). LigPrep (LigPrep, Maestro 11.2.014, Schrödinger LLC) was utilized to construct ligands, produce stereoisomers, and generate tautomers using the OPLS46 force field. Additionally, the Epik module was employed to desalt and protonate the ligands at pH 7.0. Default values were applied to the remaining parameters, and LigPrep was also used to prepare the ligands with the option 'retain specified chiralities' engaged.

In terms of ligand state assessment, Epik can calculate a penalty to quantify the energy cost of forming each state in solution based on the Hammett and Taft methods. This EpiK state penalty is entirely compatible with the GlideScore used for docking, as it is computed in kcal/mol units. This compatibility facilitates the examination of how the EpiK state penalty affects the GlideScore. The DockingScore in Glide represents the total of the GlideScore and the Epik state penalty and serves as the basis for final ranking and enrichment evaluation. Furthermore, Epik offers a technique for handling metal binding states, involving a change in the pH range within which the state is generated. This approach underscores the comprehensive considerations made in preparing the ligands for docking and subsequent analyses.

## Grid generation and molecular docking

The receptor grid complex of 6UWZ.pdb was generated after removing alpha-bungarotoxin from the binding pocket using Glide (Glide, Maestro 11.2.014, Schrödinger LLC). A virtual grid box with dimensions 20Å × 20Å × 20 Å was formed, centered within the C and F loops in the $\alpha-\delta$ binding site of the energy-minimized resting state. This grid box was used to generate a receptor grid, and the default values were assigned for other parameters. Docking was performed using the ligand docking wizard built into the Glide module of Schrödinger, with the default scaling factor and partial charge cutoff set to 0.80 and 0.15, respectively. The docking procedure consisted of two stages, extra precision (XP) followed by post-processing with prime MM-GBSA. The best dock poses were selected based on dock score and MM-GBSA energy for further investigation.

## Molecular dynamics simulations

MD simulations were conducted using the Amber22 software suite within the Amber module as detailed elsewhere (*Singh et al., 2021*; *Srivastava et al., 2022*). Ligand topology was prepared via the Leap module in AmberTools22, employing force field ff19SB (*Tian et al., 2020*) and the Generalized Amber Force Field (GAFF) (*Vassetti et al., 2019*). The AM1BCC strategy and GAFF2 were used to assign atomic charges and additional parameters. Before the simulation, the systems were minimized and equilibrated following a five-step minimization process, with each step involving 10,000 energy minimization steps. Systems were heated under NVT ensembles in two consecutive runs from 0 to 300 K for 1 ns each, followed by a 1 ns simulation at 300 K and 1 bar pressure under the NPT ensemble for equilibration. An additional 5 ns equilibration was performed before the 200 ns production MD for each system, conducted under periodic boundary conditions (PBCs) and utilizing an NPT ensemble with 300 K and 1 bar pressure. Long-range electrostatic interactions were treated using the particle mesh Ewald method (PME), including a direct space cutoff of 12.0 Å and van der Waals interactions. In production, MD, a time step of 2.0 fs was employed, and the SHAKE algorithm was used to keep bond lengths at equilibrium. Isobaric (NPT) conditions were maintained using the Berendsen barostat, with temperature monitored by a Langevin thermostat. Coordinates from the production MD were recorded in the trajectory file every 20 ps, resulting in a minimum of 10,000 frames. Trajectories were analyzed using amber-tools CPPTRAJ (*Roe and Cheatham, 2013*), and further analysis and figure generation were completed with VMD, Pymol, Xmgrace, and ChimeraX. To ensure reproducibility, MD trajectories were generated in triplicate.

## Cluster analysis

Cluster analysis of the ligand was performed using TCL scripts within the Visual Molecular Dynamics (VMD) software, as described elsewhere (*Mittal et al., 2021b*). In brief, this script automates the identification of frames from the most stable conformations based on PCA minima. Using these frames as input, the algorithm employed RMSD values of ligand positions to cluster similar orientations. Specifically, clusters with an RMSD of ≤1 Å were considered to share a similar ligand orientation. Frames that did not fit into these top clusters were automatically excluded by the script.

## PCA and free energy landscape (FEL)

PCA identifies collective motions in atomic-level simulations of macromolecules (*Mittal et al., 2021a*; *Srivastava et al., 2022*). It emphasizes significant patterns by reducing noise in MD simulations, revealing essential dynamics underlying conformational changes. All equilibrated MD simulation trajectories were analyzed by PCA computations using the CPPTRAJ algorithm using Amber tools. Prior to PCA, the least-square fitting of the protein's Cα atoms was carried out with respect to a reference structure (average protein coordinates) to eliminate rotational and translational degrees of freedom in the simulation box. A positional covariance matrix C (of dimensions $3N \times 3N$) was constructed using the Cα atom coordinates, and this matrix was diagonalized to produce eigenvectors (directions of motion) and eigenvalues (magnitudes). The elements of the covariance matrix C were obtained using:

$$C_i = \langle \left(X_i - \langle X_i \rangle\right) \left(X_j - \langle X_j \rangle\right)\rangle \ (i, j = 1, 2, 3 \ldots\ldots, 3N)$$

where $X_i$ and $X_j$ are the Cartesian coordinates of the Cα atoms, and N is the number of Cα atoms. The $\langle \rangle$ notation represents the ensemble average of atomic positions in Cartesian space. The resulting principal components (eigenvectors) were sorted by the total motion they captured, generating 3 N eigenvectors. The free energy landscapes (FELs; *Figure 3*) were created using the 'g_sham' module in GROMACS, based on the probability distribution of the top two principal components (PCs). The FEL plot aids in visualizing possible conformations and corresponding energy levels. The free energy values (G) were calculated as

$$G_i = -kT \ln \left(N_i/N_{max}\right)$$

where *k* is the Boltzmann constant, *T* is the absolute temperature of the simulated system (300 K), $N_i$ is the population of bin *i*, and $N_{max}$ is the population of the most-populated bin.

## BPMD

BPMD was performed using the Desmond module in Schrodinger, involving 10 independent meta dynamics simulations of 10 ns each to improve statistical reliability, with results averaged over the simulations. The collective variable (CV) was the RMSD of the ligand-heavy atoms relative to their starting positions, evaluated after superimposing the binding sites to account for drift. The hill height and width were set at 0.05 kcal/mol and 0.02 Å, respectively. The system was solvated in a 12.0 Å box and underwent several minimizations to slowly reach a temperature of 300 K, releasing any bad contacts or strain in the initial structure. Stability was assessed in terms of ligand RMSD fluctuations (PoseScore) and the average persistence of important contacts, such as hydrogen bonds and pi-pi interactions, between the ligand and protein residues (PersScore). Higher PersScore values indicate more stable complexes, while lower PoseScore values indicate more stable ligand positions. The CompositeScore (CompScore) combines these metrics, accounting for both ligand drift and the persistence of protein/ligand hydrogen bonds. Lower CompScore values correspond to more stable protein/ligand complexes.

## Binding free energy calculation

Experimental binding free energies were estimated by using electrophysiology, as described elsewhere (*Indurthi and Auerbach, 2023*). Briefly, the constants $K_L$ and $L_2$ (*Figure 1B*) were estimated from single-channels kinetics or dose-response curves, and $K_H$ was calculated using the thermodynamic cycle with $L_0$ known a priori. The logs of $K_L$ and $K_H$ are proportional to $\Delta G_L$ and $\Delta G_H$. The binding free energy of the protein-ligand docked complex ($\Delta G_{bind}$) was calculated in two stages, after docking (using MM-GBSA in the Prime module of Schrödinger) (*Mittal et al., 2021b*; *Purushotham et al., 2022*) and after MD simulation on equilibrated MD trajectories (using MM-PBSA/-GBSA in AMBER22). MM-PBSA is an approximation that can be useful for comparing binding affinities between different states and among agonists (*Kollman et al., 2000*; *Rastelli et al., 2010*; *Sun et al., 2014*). In our use, MM-PBSA computations included both enthalpic (ΔH) contributions from molecular interactions (van der Waals, electrostatics, solvation) and entropic (TΔS) contributions calculated using normal mode analysis. By capturing the dynamic interplay between the ligand and receptor as well as the influence

of the surrounding solvent environment this approach enables computation of a Gibbs free energy ($\Delta G$).

After docking, for the static docked pose of the protein-ligand complex, the MM-GBSA binding energy calculation was performed using the OPLS4 force field and the variable-dielectric generalized Born (VSGB) model in the Prime MM-GBSA module in *Schrödinger, 2022*; *Schrödinger, 2022*. We used $\Delta G_{bind\_dock}$ to rank protein-ligand docked complexes, selecting those with the highest negative values and dock scores as the best ligands for further studies. After MD Simulation, in the MD equilibrated system binding energies were calculated through the MM-PBSA/-GBSA protocol, implemented in AMBER22. This was performed over the most stable clustered poses extracted from the equilibrated trajectory at minima m1, m2, and m3 on the FEL over a 200 ns time frame. AMBER utilizes the conventional g_mmpbsa module for MM-PBSA calculations using the MM-PBSA.py script. These methods calculate the energies of electrostatic interactions, van der Waals interactions, polar solvation, and non-polar solvation in the equilibrated trajectory using a single trajectory protocol (STP). The binding energy of each protein-ligand complex was calculated as $\Delta Gbind = \Delta Gcomplex - \Delta Gprotein - \Delta Gligand$, where $\Delta G_{complex}$, $\Delta G_{protein}$, and $\Delta G_{ligand}$ represent the absolute free energies of the complex, protein, and ligand systems, respectively. Additional terms, such as $\Delta H$ (enthalpy), T and $\Delta S$ (temperature and total solute entropy), $\Delta G_{gas}$ (total gas phase energy), and $\Delta G_{solv}$ (solvation energy), contribute to the calculation.

The Poisson-Boltzmann calculations were performed using an internal PBSA solver in Sander, while Generalized Born ESURF was calculated using 'LCPO' surface areas. The vibrational mode entropy contributions (T$\Delta S$) for protein-ligand interactions were calculated and averaged using normal-mode analysis in Amber. The entropy term provides insights into the disorder within the system, as well as how this disorder changes during the binding process. This value was added to the enthalpy term computed by the MM-PBSA method to determine the absolute binding free energy ($\Delta G_{bind}$). These comprehensive calculations were used only as fingerprints for associating structures in simulations with states defined by experimental (electrophysiology) free energy measurement (*Figure 4A*).

## Pocket volume calculation

To investigate the variations in binding site volume across all four agonist-bound cases, pocket volume analysis was carried out using POVME3.0 (*Wagner et al., 2017*). From each of the four systems, frames were extracted at consistent intervals with a stride of 2.0, culminating in representative sets of 5000 protein structures for every system. Before commencing the volume calculation, the trajectory was aligned, and frames were selected from VMD, serving as the initial input for this method. Subsequently, an inclusion region was defined, encapsulating all binding pocket conformations within the trajectory. This region's definition involved building a sphere upon which the inclusion grid-points were computed, using different atoms of the ligand for this calculation.

## Additional information

### Funding

| Funder | Grant reference number | Author |
|---|---|---|
| NIH Blueprint for Neuroscience Research | | Anthony Auerbach |

The funders had no role in study design, data collection and interpretation, or the decision to submit the work for publication.

### Author contributions

Mrityunjay Singh, Data curation, Formal analysis, Validation, Methodology; Dinesh C Indurthi, Conceptualization, Supervision, Investigation, Writing - original draft, Project administration, Writing – review and editing; Lovika Mittal, Data curation, Formal analysis; Anthony Auerbach, Conceptualization, Resources, Supervision, Project administration, Writing – review and editing; Shailendra Asthana, Formal analysis, Supervision, Investigation, Methodology, Project administration, Writing – review and editing

## Author ORCIDs
Dinesh C Indurthi https://orcid.org/0000-0001-8837-5883
Anthony Auerbach https://orcid.org/0000-0003-4151-860X

Reviewer #1 (Public Review): https://doi.org/10.7554/eLife.92418.4.sa1
Reviewer #2 (Public Review): https://doi.org/10.7554/eLife.92418.4.sa2
Reviewer #3 (Public Review): https://doi.org/10.7554/eLife.92418.4.sa3
Reviewer #4 (Public Review): https://doi.org/10.7554/eLife.92418.4.sa4
Author response https://doi.org/10.7554/eLife.92418.4.sa5

## Additional files

### Supplementary files
- Supplementary file 1. Supplementary material.
- MDAR checklist

### Data availability
All data generated or analysed during this study are included in the manuscript and supporting files; source data files have been provided for Figures 2, 3 and 4.

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
