## [Editor Report · eLife Assessment]

This **useful** work provides insight into agonist binding to nicotinic acetylcholine receptors, which is the stimulus for channel activation that regulates muscle contraction at the neuromuscular junction. The authors use in silico methods to explore the transient conformational change from a low to high affinity agonist-bound conformation as occurs during channel opening, but for which structural information is lacking owing to its transient nature. The simulations indicating that ligands flip ~180 degrees in the binding site as it transitions from a low to high affinity bound conformation are **solid**. A limitation is the approximation of binding energies using Poisson-Boltzmann Surface Area and mismatch between calculated and experimental binding energies for two of the four ligands tested. Nonetheless, this work presents an intriguing picture for the nature of a transient conformational change at the agonist binding site correlated with channel opening.

---

## [Referee Report · Reviewer #1 (Public Review)]

Summary:

The authors want to understand fundamental steps in ligand binding to muscle nicotinic receptors using computational methods. Overall, although the work provides new information and support for existing models of ligand activation of this receptor type, some limitations in the methods and approach mean that the findings are not as conclusive as hoped.

Strengths:

The strengths include the number of ligands tried, and the comparison to the existing mature analysis of receptor function from the senior author's lab.

Weaknesses:

The weakness are the brevity of the simulations, the concomitant lack of scope of the simulations, the lack of depth in the analysis and the incomplete relation to other relevant work. The free energy methods used seem to lack accuracy - they are only correct for 2 out of 4 ligands.

---

## [Referee Report · Reviewer #2 (Public Review)]

Summary:

The aim of this manuscript is to use molecular dynamics (MD) simulations to describe the conformational changes of the neurotransmitter binding site of a nicotinic receptor. The study uses a simplified model including the alpha-delta subunit interface of the extracellular domain of the channel and describes the binding of four agonists to observe conformational changes during the weak to strong affinity transition.

Strength:

The 200 ns-long simulations of this model suggest that the agonist rotates about its centre in a 'flip' motion, while loop C 'flops' to restructure the site. The changes appear to be reproduced across simulations and different ligands and are thus a strong point of the study.

Weaknesses:

After carrying out all-atom molecular dynamics, the authors revert to a model of binding using continuum Poisson-Boltzmann, surface area and vibrational entropy. The motivations for and limitations associated with this approximate model for the thermodynamics of binding, rather than using modern atomistic MD free energy methods (that would fully incorporate configurational sampling of the protein, ligand and solvent) could be provided. Despite this, the authors report correlation between their free energy estimates and those inferred from the experiment. This did, however, reveal shortcomings for two of the agonists. The authors mention their trouble getting correlation to experiment for Ebt and Ebx and refer to up to 130% errors in free energy. But this is far worse than a simple proportional error, because -24 Vs -10 kcal/mol is a massive overestimation of free energy, as would be evident if it the authors were to instead to express results in terms of KD values (which would have error exceeding a billion fold). The MD analysis could be improved with better measures of convergence, as well as a more careful discussion of free energy maps as function of identified principal components, as described below. Overall, however, the study has provided useful observations and interpretations of agonist binding that will help understand pentameric ligand-gated ion channel activation.

---

## [Referee Report · Reviewer #3 (Public Review)]

Summary:

The authors use docking and molecular dynamics (MD) simulations to investigate transient conformations that are otherwise difficult to resolve experimentally. The docking and simulations suggest an interesting series of events whereby agonists initially bind to the low affinity site and then flip 180 degrees as the site contracts to its high affinity conformation. This work will be of interest to the ion channel community and to biophysical studies of pentameric ligand-gated channels.

Strengths:

I find the premise for the simulations to be good, starting with an antagonist bound structure as an estimate of the low affinity binding site conformation, then docking agonists into the site and using MD to allow the site to relax to a higher affinity conformation that is similar to structures in complex with agonists. The predictions are interesting and provide a view into what a transient conformation that is difficult to observe experimentally might be like.

Weaknesses:

A weakness is that the relevance of the initial docked low affinity orientations depend solely on in silco results, for which simulated vs experimental binding energies deviate substantially for two of the four ligands tested. This raises some doubt as to the validity of the simulations. I acknowledge that the calculated binding energies for two of the ligands were closer to experiment, and simulated efficiencies were a good representation of experimental measures, which gives some support to the relevance of the in silico observations. Regardless, some of the reviewers comments regarding the simulation methodology were not seriously addressed.

---

## [Referee Report · Reviewer #4 (Public Review)]

Summary:

In their revised manuscript "Conformational dynamics of a nicotinic receptor neurotransmitter binding site," Singh and colleagues present molecular docking and dynamics simulations to explore the initial conformational changes associated with agonist binding in the muscle nicotinic acetylcholine receptor, in context with the extensive experimental literature on this system. Their central findings are of a consistently preferred pose for agonists upon initial association with a resting channel, followed by a dramatic rotation of the ligand and contraction of a critical loop over the binding site. Principal component analysis also suggests the formation of an intermediate complex, not yet captured in structural studies. Binding free energy estimates are consistent with the evolution of a higher-affinity complex following agonist binding, with a ligand efficiency notably similar to experimental values. Snapshot comparisons provide a structural rationale for these changes on the basis of pocket volume, hydration, and rearrangement of key residues at the subunit interface.

Strengths:

Docking results are clearly presented and remarkably consistent. Simulations are produced in triplicate with each of four different agonists, providing an informative basis for internal validation. They identify an intriguing transition in ligand pose, not well documented in experimental structures, and potentially applicable to mechanistic or even pharmacological modeling of this and related receptor systems. The paper seems a notable example of integrating quantitative structure-function analysis with systematic computational modeling and simulations, likely applicable to the wider journal audience.

Weaknesses:

The response to the initial review is somewhat disappointing, declining in some places to implement suggested clarifications, and propagating apparent errors in at least one table (Fig 2-source data 1). Some legends (e.g. Fig 2-supplement 4, Fig 3, Fig 4) and figure shadings (e.g. Fig 2-supplement 2, Fig 6-supplement 2) remain unclear. Apparent convergence of agonist-docked simulations towards a desensitized state (l 184) is difficult to interpret in absence of comparative values with other states, systems, etc. In more general concerns, aside from the limited timescales (200 ns) that do not capture global rearrangements, it is not obvious that landscapes constructed on two principal components to identify endpoint and intermediate states (Fig 3) are highly robust or reproducible, nor whether they relate consistently to experimental structures.

---

## [Author Response]

The following is the authors’ response to the previous reviews.

The Editors have assessed your revised submission and rather than issuing a further decision letter we are writing to invite you to make a few small amendments to this version of the paper as listed below.

We added a summary paragraph at the end of the introduction for clarity.

(1) RMSD values in Fig 2-source data 1 (and possibly reflected in Fig 2C) appear to be improbably duplicated, specifically ACh runs 1/2, Ebx runs 1/3, and error values for Ebx vs. ACh.

Thanks for bringing this to our attention. The values are now corrected.

(2) Shaded area in Fig 2-supplement 5D is inaccurate for depicting loop C.

The shaded area now reflects residues in loop C, residues 189-198.

(3) In Fig 2-supplement 4 where an abrupt change in ligand RMSD is implied to represent a cis-trans flip, the accompanying figure showing snapshots misleadingly depicts a different simulation of CCh instead of ACh.

The snapshot was from the correct ACh simulation. It was mislabeled as CCh in the legend, which now stands corrected.

(4) Legend to Fig 3 seems misleading regarding colors in the porcupine plots.

The color pattern indicated in the legend represents the FEL plot and not the porcupine plot. Description about the porcupine plot is not associated with any color.

(5) Some shaded regions in Fig 6-supplement 2 do not correspond to intervals reported in Fig 4-source data 1.

Thanks. This is now corrected to match the table.

Given that some of the above points have remained unaddressed from the prior round of review, the authors should double check that they have addressed any other relevant prior comments not explicitly listed here.Finally, the revised first results section has removed the explanation as to why the authors opted to simulate a dimer (i.e., affinity being affected only by local perturbations). The authors should consider reincorporating this explanation for readers, as well as adding a reference to Wang et al. 1997 (PMID: 9222901) in regard to lines 116-119.

The revised section now includes an added explanation on why dimer was used in simulations. Gupta et. al., J Gen Physiol. 2017 Jan; 149(1): 85–103 was added, as it includes residues from not just the M1 domain that Wang et al covers, but other TMD regions also.